# Laugh before You Study: Does Watching Funny Videos before Study Facilitate Learning?

**DOI:** 10.3390/ijerph19084434

**Published:** 2022-04-07

**Authors:** Mengke Wang, Zengzhao Chen

**Affiliations:** Faculty of Artificial Intelligence in Education, Central China Normal University, Wuhan 430079, China; moco@mails.ccnu.edu.cn

**Keywords:** pre-learning emotion, funny videos, positive emotion, cognitive-affective model, multimedia learning, emotion induction

## Abstract

Emotions exist widely in the entire process of learning and affect students’ motivation as well as academic performance. In multimedia learning, academics usually focus on the impact of teachers’ emotions or the emotional design of multimedia learning materials on students’ emotions and learning results. Few studies have investigated how to enhance learning by regulating students’ pre-learning emotions. This study focused on whether playing funny videos before learning could promote students’ positive emotions to enhance their motivation, satisfaction, and learning outcomes. We randomly divided 81 elementary school students into two groups: experimental group and control group. While the experimental group watched funny video clips, the control group watched neutral video clips before starting the video learning. The experimental group had more positive pre-learning emotions than the control group. After the course, the emotion of the experimental group declined while that of the control group enhanced. However, positive pre-learning emotions still promoted students’ understanding and transfer of learning materials. Moreover, no significant differences were observed between the two groups in learning motivation, satisfaction, and retention tests. Furthermore, this paper analyzed the causes of the experimental results and discussed the insights for teaching.

## 1. Introduction

Emotion is a key factor affecting the process and outcome of learning. Academic settings are abound with emotions such as happiness, enjoyment, hope, pride, anger, anxiety, shame, hope, or boredom. The study of emotion in learning has aroused the interest of many researchers. Many studies have shown that some emotions promote learning while some hinder it. Based on how they promote or hinder learning, emotions can be divided into two broad categories, positive and negative. Happiness, enjoyment, and hope are examples of positive emotions that can boost intrinsic and extrinsic motivation, enhance the use of flexible learning strategies, and assist self-regulation, all of which can improve academic performance under most circumstances [1]. Negative emotions such as boredom and anger impede students’ motivation and learning engagement, thereby hinder learning [2]. Hence, how to arouse students’ positive learning emotions is an important prerequisite to achieve the desired learning results.

In the study of emotion in multimedia learning, previous studies have mainly revealed that the positive emotions of instructors in video lectures and the emotional design of learning materials can promote the generation of positive emotions, so as to promote students to achieve better learning results. These studies provide empirical evidence for the integrated cognitive-affective theory of learning with multimedia. However, promoting students’ learning through external emotional induction has not been fully studied. Emotions are dynamic and not as stable as mood or affect [3]; they are easily triggered and affected. In multimedia learning, students’ emotions can be influenced by three sources. First, learners’ own emotions. Students do not start learning without emotions, and their pre-learning emotions are inherent just as their prior knowledge, but they are not as stable and lasting as prior knowledge. The second is the emotion generated by the interaction with the learning material. Shapes, colors, pictures, and emotional attributes of learning materials affect emotions. For example, bright colors tend to give people a bright feeling and make them feel positive, while gray tends to give people a dull feeling and make them feel sad [4]. The third situation involves an instructional video with the teacher’s presence. The instructor’s vocal feedback, facial expressions, tone of voice, and body posture all convey emotions. From the existing literature, most studies have focused on the influence of the latter two emotional sources on students’ learning, while inadequate attention has been paid to the first emotional source—pre-learning emotion.

Students’ pre-learning emotion can be induced by various external materials. We often see students viewing funny videos to relieve pressure and regulate emotions while learning. It remains unclear whether funny videos are effective in inducing positive emotion and boosting learning, or how watching funny videos before learning changes students’ emotion after learning. Therefore, this study aimed to investigate how learners’ pre-learning emotions affect learning outcomes by determining whether the use of funny videos before learning evokes positive emotions and facilitates learning. Learning results were assessed by a retention and a transfer test. Moreover, measures of emotions after learning as well as motivation and satisfaction were used to elucidate how the emotional state might affect learning.

## 2. Literature Review

### 2.1. Emotion and Multimedia Learning

Emotions are intrinsically motivating and intertwined with cognition, allowing and sustained cognitive activity including key learning mechanisms such as attention and memory [5]. The integrated cognitive-affective theory of learning with media stipulates that emotions and cognition are inseparable in multimedia learning. Learners experience emotional responses when they perceive auditory and visual information in the learning environment. In the working memory phase, emotions are involved in the choice of texts and images of the material process; in this stage, emotion is a type of mood. In the organization stage, emotions are involved in the visual process of psychological representation of image materials, with interest and motivation influencing each other. Finally, after the integration stage, the emotional schema is formed and stored in the long-term memory.

Considering the key role of emotions in multimedia learning, it has become a research hotspot to influence learners’ emotional experience and thus learning results through emotional design of learning materials. Lately, a growing number of studies have focused on emotional design of educational videos that support learning. For example, Mayer and Estrella [6] redrew the learning material of how a virus causes colds using bright colors such as red or blue, with expressive eyes (registering surprise, fear, and sickness at various stages in the process). The results revealed that the enhanced emotional design markedly improved students’ learning performance compared with the black-and-white design. A more detailed comparison of appearance elements is provided in Uzun and Yıldırım [7]. Not limited to comparing colors and anthropomorphizing features, they added the element of sound. Specifically, they designed teaching materials with four different levels of emotional design: neutral, color, anthropomorphic, and anthropomorphic plus sound. The results revealed that positive emotions typically increased as the amount of emotional design features increased. Besides the shapes, colors, and anthropomorphic features of multimedia elements, the text exhibited an emotional potential, which could be estimated according to the linguistic aspects on how many emotional aspects (e.g., emotional states and emotional situations) are expressed in the text. Stark et al. [8] enriched the original text with either positive or negative emotional parentheses to test its impact on learning, establishing that the emotional text designs improved cognitive processing. Although these studies confirmed the impact of emotional design on learning, some studies highlighted that students’ emotions actually decreased after learning emotional learning materials. That is, positive emotional design did not increase the learning effect by influencing students’ emotions but promoted more effective information selection, organization, and processing by bright colors or eye-catching shapes, thereby improving students’ performance in learning tasks [9].

In video lectures with the instructor’s presence, the instructor’s emotion also attracted many researchers’ attention. To promote positive emotional states among students, some studies focused on the instructor’s emotion expression. For example, Lawson et al. [10] explored whether students noticed an instructor’s emotions during an instructional video, as well as how effectively participants could perceive different emotions depicted by a human and virtual instructor (i.e., animated pedagogical agent) in a video lecture. The results revealed that participants could recognize each of the emotions displayed by the instructor. Furthermore, without the presence of the teacher image, learners could distinguish whether the teacher was in a positive or negative emotion just by listening to the teacher’s voice [11]. Moreover, students rated the teacher with a positive voice higher. Regarding learning outcomes, students who studied from teachers with positive emotions attained better learning outcomes than those who studied from teachers with negative and neutral emotions [12]. In conclusion, the above mentioned studies demonstrated that instructors’ positive emotions can promote students’ learning owing to the impact of teachers’ emotions on students’ emotions. Students’ emotions become positive when they learned from teachers with positive emotions, which in turn stimulated students’ learning engagement through better learning motivation, and ultimately led to better learning outcomes.

Overall, the studies mentioned above aimed to enhance the learning effect through emotional design of learning materials or teachers’ positive emotional expression in the process of learning. However, these studies overlooked students’ pre-learning emotions. Learners always have a basic emotion when entering the learning situation. The learners’ pre-learning emotions is a crucial personality characteristic that cannot be ignored in teaching, such as students’ previous knowledge and experience, and it exerts a more profound impact on the learning process, even affecting how students perceive teachers’ emotions and the emotional design of learning materials [13]. Thus, this study aimed to investigate whether promoting students’ pre-learning emotion can enhance the learning outcome.

### 2.2. Pre-Learning Emotion

Emotions are situational; that is, different emotions are triggered in different situations. Thus, the research on emotions should be combined with specific situations. According to different academic scenes, emotions were divided into class-related emotions, learning-related emotions, and test-related emotions. For each situation, it could be categorized into three parts based on the time period. For example, learning-related emotions could be divided into pre-learning emotions, during-learning emotions, and post-learning emotions. In this study, we largely focused on learning related emotions and more specifically, pre-learning emotions.

To date, few empirical studies have focused on learners’ pre-learning emotions, and only two studies have investigated this issue. Park et al. [14] established that by inducing students to have positive pre-learning emotions, their understanding of learning content and transfer performance could be improved; however, they did not compare how negative pre-learning emotions affected learning outcomes. Knörzer et al. [15] filled this gap by using a combination of music and autobiographic recall to induce learners to produce positive, negative, or neutral emotions in the laboratory environment. The negative emotion group outperformed the positive emotion group, according to the findings. However, they claimed that they could not deduce whether the negative emotional state before learning was beneficial to learning because the negative emotion group was actually in a neutral emotional state in the experiment, and the results should be explained such that the positive emotion exerted an adverse impact on learning. In other studies, the highly active positive emotional state broadened students’ scope of attention [16] or distract students from learning materials [17]. The negative or neutral emotional state made students’ attention more focused and cognitive processing more detailed. Although the conclusions of the two studies mentioned above are inconsistent, this might be caused by the sample and task difficulty; however, both showed that learners’ pre-learning emotions are essential factors that affect cognitive results.

### 2.3. Emotion Induction

Emotion induction and emotion regulation are similar but different terms; both denote the change or influence of emotion. The difference between the two is that emotion induction is primarily influenced by others by certain means and methods. Emotion regulation includes not only the influence and change of others on individual emotions but also the influence and change of oneself on emotions [18]. In this study, we mainly referred to the emotional change of individuals by others, that is, emotion induction. In addition, emotions are the psychological process evoked by a perception of an event, a memory, and specific types of media such as photographs, voice, and words [19]. Siedlecka and Denson [20] broadly classified emotion induction techniques into five specific methods: visual stimuli (including static images or videos), listening to music, autobiographical recall, situational procedures, and imagery.

When it comes to inducing students’ pre-learning emotions, video may be the best choice. In traditional classroom teaching, we often see or experience that teachers organize students to watch a video before class to stimulate their positive emotions. Learners themselves often unconsciously use short funny videos to regulate their emotions; they watch short videos for entertainment and relaxation after studying for a period, and then re-devote themselves to their studies. In addition to its good ecological validity, video as an emotion inducing material has the following advantages. First, videos convey more information than pictures or sounds alone and thus induce emotions quicker. Second, unlike autobiographical recall and imaginary which relies on individual personality traits, video presents external information to elicit emotion; thus, it induces cleaner emotions [21]. Third, videos save time and manpower when compared with situational procedures, which require creating a social scenario that elicit the desired emotion. Overall, using videos to induce students’ emotions is a relatively simple, effective, and scalable method.

### 2.4. Research Questions

Can watching funny videos before study improve learning? This study explored whether using funny videos to induce students’ pre-learning emotions can effectively promote their learning. Although two previous studies [14,15] have examined the effects of regulating learners’ pre-learning emotions on multimedia learning outcomes, participants learned the material independent of their professional background. In addition, the researchers highlighted that future research should focus on whether inducing students’ pre-learning emotions can produce different results when learning materials related to their own learning task; that is, in a conventional learning environment, students’ motivation to learn is different from participating in an experiment to learn unrelated learning materials. Motivation is a key driver of learning; therefore, it might lead to conclusions different from those of existing studies. This study explored whether regulating students’ pre-learning emotions while learning materials related to their own tasks would affect their learning outcomes. In addition, unlike previous two studies, participants in our experiment were primary school students, and we used funny videos as a means of emotion induction. 

The specific research questions addressed in this study are:
RQ1: Can funny videos make students in the experimental group feel more positive before learning?RQ2: How is the change of emotion between the two groups after learning?RQ3: Does the experimental group have higher learning motivation and satisfaction than the control group?RQ4: Does the experimental group perform better in knowledge retention and knowledge transfer than the control group?

## 3. Materials and Methods

### 3.1. Participants and Experimental Design

In this study, participants were 5th grade (approx. age: 11 years) students from a primary school in south-central China. A total of 81 students participated in the study. The students here had video learning experience; they often used computers and iPads to surf the Internet to collect learning materials or view videos at school and at home. All participants gave written informed consent of their parents and received a small gift for participating in the study. The Academic Committee of the School of Psychology at CCNU approved the study protocol.

### 3.2. Materials

#### 3.2.1. Emotion Induction Material

We edited two short videos (2–3 min) to induce positive emotions as well as neutral emotions. One video was excerpts from an online program named “light moment”, with a child and a dog eating lunch together to induce positive emotions (Figure 1). The other was a multicolored display of moving blue bars to induce neutral emotions (Figure 2).

#### 3.2.2. Learning Material

We created an instructional video, the teaching content of which was the elementary science knowledge point “The Composition of Rocks”. In order to reduce the influence of emotional design of learning materials on students’ emotion in the process of learning, we strictly controlled the emotional design of various multimedia elements to make them present neutral emotional colors. Specifically, the video manuscript, as well as the text and pictures in the slides all did not contain strong emotional attributes. As for the instructor, she kept a gentle facial expression, fluctuated her tone, and faced her body toward the camera. When she used gestures to point to the teaching material, her eyes also looked at the teaching material, and her body turned to the teaching material. (Figure 3). We also invited six master-level graduate students and two doctoral students majoring in education to watch the video materials and score the emotional attributes of the video materials. Among them, 1 represents very negative, 4 represents neutral, and 7 represents very positive. The results showed that the affective attribute of learning materials was neutral (M = 4.25, SD = 0.43).

### 3.3. Measures

The measurements included one demographic questionnaire, three knowledge tests, two emotion questionnaires, one learning motivation scale, and one learning satisfaction scale.

#### 3.3.1. Demographic Questionnaire

Participants were asked to report their sex, age, and year in school.

#### 3.3.2. Knowledge Tests

We invited a science expert, together with the teacher in the video lecture, to design three knowledge tests: (I) a prior knowledge test; (II) a knowledge retention test; and (III) a knowledge transfer test. The pre-knowledge test was designed to understand students’ background knowledge about rocks. It included true or false questions (e.g., gold is not a mineral), single-choice questions (e.g., what is true about the composition of rocks? (A) rock is composed of one mineral, (B) rock is composed of three or more minerals, and (C) rock is composed of one or more minerals), and multiple-choice questions (e.g., the following characteristics of rocks are? (A) color, (B) length, and (C) gloss), with a total of 10 points; the higher the score, the higher the level of knowledge. The retention test was designed to measure students’ memory of the content. The answers to the questions could be found directly from the video material. It included fill-in-the-blank questions (e.g., the hardest mineral found in nature is ____), true or false questions (e.g., sand is a kind of rock), and picture recognition questions (the pictures included in this section can be found in Appendix A), with a total of 18 points; the higher the score, the better the knowledge retention. The transfer test was designed to test students’ understanding of the content and its application to the knowledge [22]. The answers to the questions could not be found directly from the video materials. It required learners to make appropriate reasoning based on their understanding of the learning material. It included true or false questions (e.g., rocks and minerals are the mineral resources of the earth, not the resources that people produce and live in), single choice questions (e.g., the reflection of light on the surface of rock forms the ___ of rock. (A) color, (B) gloss, and (C) transparency), and a short answer question (e.g., name two examples of the use of rocks in your life), with a total of 13 points; the higher the score, the better the knowledge transfer.

#### 3.3.3. Emotion Questionnaires

We adopted the emotion questionnaires from Horovitz and Mayer [23] to evaluate students’ pre-learning emotions and post-learning emotions.

Before learning, participants rated their emotions on a five-point Likert scale (1 = strongly disagree and 5 = strongly agree) on the following four items: before learning, I feel happy; before learning, I feel content; before learning, I feel frustrated; before learning, I feel bored. After class, participants rated their emotions. Students’ post-class emotions also included four questions: after learning, I feel happy; after learning, I feel content. after learning, I feel frustrated. after learning, I feel bored; these four questions used the five-point Likert scale (1 = strongly disagree and 5 = strongly agree).

#### 3.3.4. Learning Motivation

The learning motivation questionnaire was an excerpt from Stull et al. [24], containing six items on participants’ enjoyment, willingness to learn in this way in the future, understanding of the learning materials, desire to learn more about the content, finding the lesson useful, and motivation to learn the content. The questionnaire used a seven-point Likert scale (1 = strongly disagree and 7 = strongly agree). In this study, Cronbach *α* for learning motivation was 0.904.

#### 3.3.5. Learning Satisfaction

The learning satisfaction questionnaire contained three items to measure students’ satisfaction with the teacher’s teaching, teaching content, and learning environment. Each item used a five-point Likert scale (1 = strongly disagree and 5 = strongly agree). In this study, Cronbach *α* for learning satisfaction was 0.814.

### 3.4. Procedure

This study was conducted in a multimedia classroom with a projector. All participants came from two parallel classes, that is, there was no significant difference in the students’ academic performance in the two classes. Moreover, the two classes had similar sex ratio and age distribution. Therefore, the randomization process was class-based. We randomly assigned one class as the control group and the other as the experimental group. In the control group, there were 22 boys and 18 girls. In the experimental group, there were 23 boys and 18 girls. Both groups were escorted into different multimedia classroom where the study was conducted. The order of measurement implementation may affects the accuracy of the measurement [25]; we fully considered this point. The specific procedure was shown as below:
First, all participants filled out the demographic questionnaire and prior knowledge test. Then, students in the control group watched blue moving bars for 2 min. Meanwhile, students in the experimental group watched funny videos for about 2 min.Second, the control group and the experimental group filled in the pre-learning emotion questionnaire. Rather than taking an emotion test before the prior knowledge test, we avoided pre-learning emotions reported by students from being affected by the prior knowledge.Third, both groups watched the same video to learn the material.Fourth, both groups reported their post-learning emotions.Fifth, both groups did retention and transfer tests.Finally, students were invited to report their motivation and satisfaction with the course.


### 3.5. Statistical Analysis

Different data analysis methods were used in this study to address the research question. For RQ1, RQ3, and RQ4, the independent sample *t*-test was conducted to compare the differences between the experimental group and the control group on several dependent variables. Our null hypothesis H0 states that there was no significant difference between the experimental group and the control group in pre-learning emotion, motivation, satisfaction, retention score, and transfer score. The formula is μ_1_ − μ_2_ = 0; accordingly, our alternative hypothesis H0 states that there was a significant difference between the experimental group and the control group in pre-learning emotion, motivation, satisfaction, retention score, and transfer score. The formula is μ_1_ − μ_2_ ≠ 0. If the significance of *t* is less than 0.05, the null hypothesis should be rejected; otherwise, the null hypothesis should be accepted. *p* value was considered statistically significant when it was two-tailed.

For RQ2, the generalized estimation equation (GEE) was used to compare the emotional differences between the two groups at different time points and the emotion differences of each group at different time points. Although repeated measure ANOVA was also applicable to solve this problem, the prerequisite conditions of repeated measure ANOVA are relatively strict such as normality, homogeneity, and particularly sphericity [26]. The generalized estimation equation (GEE) has unique advantages in data analysis of repeated measurements and can support multiple data types and distribution patterns [27]. This study mainly involved two independent variables, time and group, and four dependent variables: happy, content, frustrated, and bored. To answer the research question, we analyzed the interaction effect of time and group with one emotion as the dependent variable. The model of the five correlation structures was carried out and the QIC value was recorded. The structure that obtains the smaller QIC value shows better fit of the model to the data [28]. The QIC results are shown in Table 1. It suggests that AR, exchangeable, M-dependent, and unstructured all had a smaller QIC compared to independence. Thus, we could choose one of them except independence for the correlation structure to build the model.

## 4. Results

To adequately interpret the results, it was crucial to determine whether the groups differed significantly on basic characteristics. Thus, we conducted an independent sample *t*-test between the two groups. As shown in Table 2, the independent sample *t*-test showed no significant differences in prior knowledge between the two groups (*p* = 0.703).

### 4.1. RQ1: Can Funny Videos Make Students in the Experimental Group Feel More Positive before Learning?

To examine whether the funny video promoted the pre-learning emotions of the experimental group, an independent sample *t*-test was conducted on the pre-learning emotion data of the two groups. A significant difference was noted in pre-learning emotions, as shown in Table 3. Students in the experimental group were significantly happier (*p <* 0.001) and more content (*p* = 0.002) than students in the control group. In addition, students in the control group were significantly more frustrated (*p* = 0.003) and more bored (*p* = 0.001) than students in the experimental group.

### 4.2. RQ2: How Is the Change of Emotion between Two Groups after Learning?

To examine the change in emotions between the two groups after learning, we conducted generalized estimation equations(GEE) to analyze the data. The tests of model effect results are shown in Table 4. It shows that the interaction effect of time and group was not significant for the happy and content emotions, but for the frustrated and bored emotions, the interaction effect of time and group was significant.

The estimated marginal mean and pairwise comparison results are shown in Table 5. The rows 3–6 of the table show the interaction between time and group; it represents the differences in emotion between the experimental group and the control group at different time points. It can be seen that before learning, there were significant differences in happy, content, frustrated, and bored emotions between the experimental group and the control group. Specifically, the experimental group was happier and more content than the control group. The control group was more frustrated and bored than the experimental group. After learning, there were still significant differences in happy, frustrated, and bored emotions between the experimental and control groups. The experimental group was happier than the control group, and the control group was more frustrated and bored than the experimental group. There was no significant difference in content emotion between the two groups.

The last four rows of the Table 5 show the interaction between group and time; it represents the difference in emotion between the experimental group and the control group at different time points. It can be seen that the happy, frustrated, and bored emotions of the experimental group changed significantly before and after learning. Specifically, after learning, the students’ happiness decreased and frustration and boredom increased. Students’ content emotion also decreased, but there was no significant difference. For the control group, happy, content, frustrated, and bored emotions did not change significantly before and after learning. However, it can be seen from the mean value that the happy and content emotions of the control group showed an upward trend, while the frustrated and bored showed a downward trend. The emotion changes of different groups before and after learning are illustrated in Figure 4 and Figure 5. The overall emotion changes of the two groups at different time points are shown in Figure 6.

### 4.3. RQ3: Does the Experimental Group Have Higher Learning Motivation and Satisfaction than the Control Group?

To determine any difference between the two groups on motivation and satisfaction, the independent sample *t*-test was used. The results showed no significant differences on motivation (*p* = 0.286) and satisfaction (*p* = 0.520). Table 6 presents the descriptive data.

### 4.4. RQ4: Does the Experimental Group Perform Better in Knowledge Retention and Knowledge Transfer than the Control Group?

To determine any difference between the two groups on retention and knowledge transfer, the independent sample *t*-test was used. The results showed no significant differences in knowledge retention (*p* = 0.143) between the two groups. However, the transfer test (*p* < 0.001) of the experimental group was significantly higher than the control group. Table 7 presents the descriptive data.

## 5. Discussion

### 5.1. This Work

This study examined the effect of students’ pre-learning emotions on primary students’ emotional state, motivation, satisfaction, and learning performance in video lectures. The following is an analysis and discussion of the research results.

#### 5.1.1. RQ1 Can Funny Videos Make Students in the Experimental Group Feel More Positive before Learning?

Students in the experimental group were significantly happier (*p* < 0.001) and more content (*p* = 0.002) than students in the control group. Meanwhile, students in the control group were significantly more frustrated (*p* = 0.003) and more bored (*p* = 0.001) than students in the experimental group. These findings corroborate previous studies. Abel and Maxwell [29] reported that viewing a humorous video compared with a nonhumorous video reduced anxiety and improved positive affect under both low and high stress. Moreover, when watching a funny video, one experienced joy and delight [30].

#### 5.1.2. RQ2 How Is the Change of Emotion between Two Groups after Learning?

After completing video learning, students’ emotions in the experimental group became significantly negative, while that of the control group became slightly but not significantly positive. This could be a comparison effect; that is, the comparison between the induction videos and the learning videos resulted in the different trends in emotions of both groups. Regarding the experimental group, the learning videos were not as interesting as the funny videos; thus, the students’ emotions became negative after watching the learning videos. Regarding the control group, the learning videos were less monotonous than the color-block videos; thus, after watching the learning video, students’ emotions became slightly but not significantly positive.

#### 5.1.3. RQ3 Does the Experimental Group Have Higher Learning Motivation and Satisfaction than the Control Group?

We found no significant difference in motivation and satisfaction between the two groups, which is contrary to a previous study. It was confirmed that students’ enjoyment of learning positively correlated with their intrinsic and extrinsic motivation [22], whereas correlations for boredom with motivation were negative. The lack of significant difference in motivation between both groups is attributable to the way emotions were induced. The funny video materials used in this study were funny fail videos of humans and animals. Per Gable and Harmon-Jones [31] proved that cute animal and human videos triggered emotions of low motivational intensity. Food videos, instead, stimulated high motivational intensity. Thus, the motivation of positive emotions evoked by video clips used in this study was limited. In addition, it correlated with the time of emotion induction. The duration of emotion induction before a formal learning task might not be as long as the duration of emotion induced by the learning environment. Thus, temporary positive emotional experiences might not motivate students to learn.

Regarding satisfaction, we also found no significant difference in learning satisfaction between the two groups. A previous study confirmed that emotions could influence a learner’s attitude toward something or somebody [32]. However, in this study, we did not find a significant difference between the two groups, which could be caused by the comparison, with the control group in a natural emotion rather than a negative emotion. Artino [33] demonstrated that happy emotions positively predicted students’ learning satisfaction, while depression negatively predicted students’ learning satisfaction. In addition, students’ emotions and motivation jointly influenced satisfaction, and they explained a significant portion of variance in satisfaction [34]. Owing to no significant difference in motivation and the emotion difference being small after learning between the two groups, no significant difference was found in satisfaction.

#### 5.1.4. RQ4 Does the Experimental Group Perform Better in Knowledge Retention and Knowledge Transfer than the Control Group?

We found no significant difference in the retention test, which mainly focused on students’ memory of learning material; this is attributable to no significant difference in students’ learning motivation. Motivation is the driving force behind cognitive processing, which results in improved learning outcomes. The second reason may due to the level of difficulty of retention tests. Perhaps positive emotions enhance performance on tasks that mostly require divergent thinking (e.g., think about the various uses of the pencil) [35]. In addition, the retention test largely examined the students’ knowledge through memory and repetition; it did not require much positive emotion input, but negative emotions impaired students’ memory [36]. The third reason is that the activation levels of emotions might influence performance on tasks [37]. Finally, the funny video we used in this study was not related to the learning content. A previous study demonstrated that funny videos that were compatible with the course topic boosted student acquisition and content retention [38].

Regarding the transfer test, it reflected the deep processing of learning materials. We found a significant difference between the two groups in the transfer test, and the experimental group performed better in the transfer test, which is consistent with a previous study. Um et al. [22] showed that external induction of positive emotions did not enhance retention but did improve transfer. Moreover, Politis and Houtz [39] also found that participants who in the positive emotion condition were significantly more fluent when solving creative problem than those who watched the neutral video. Gökçen showed that watching a funny video, especially about a foreign language, affected learning [40]; this may be because positive emotions improved the flexibility of thinking, promoted deep processing, and thus improved transfer performance [41]. The second reason could be that positive pre-learning emotions can increase cognitive engagement, which results in better transfer performance [42].

### 5.2. Theoretical and Practical Contributions

This study extended the literature on emotions in multimedia learning beyond college samples and further considered pre-learning emotion induction. We examined the effects of watching funny videos before learning on primary school students’ learning outcomes. This study supported the integrated cognitive-affective theory of learning with media and the understanding that positive emotions promote learners’ cognitive processing. When students are in a positive emotion, they are more willing to put more mental work into processing the learning material, thereby producing better learning results. Moreover, the study offered important new insights into emotions. Learners’ pre-learning emotions are the real starting point that affects their deep learning.

The practical implication of this study is that the efficacy of funny short videos as a regulation of learning emotions was verified. This study shed light on teaching practices in the following two aspects. First, the appropriate use of short videos can promote students’ positive emotions and thus improve their learning performance. Of note, watching funny videos is often used for leisure and entertainment, but few researchers have used it as a means of regulating students’ emotions. Teachers can evoke positive emotions of students by using this easy and economic method. Second, teachers should pay attention to students’ pre-learning emotions which affect their learning results. It is wise to execute some activities such as singing and playing games to prepare students for positive emotions before learning.

### 5.3. Limitation and Future Research

Although meaningful findings were reported in this study, there are three limitations. First, this study only used self-ratings of emotional state; it was subjective data. Although previous studies have highlighted that self-reported arousal correlates with students’ regulation of their effort in task and self-reported valence correlates with cognitive regulation processes. Our study lacked some objectivity in this regard. Thus, future research would benefit from more direct measures such as skin conductance response (SCR) and facial muscle electromyogram (EMG). With the combination of multimodal objective (SCR and EMG) and subjective (self-report) data, the learning process can be captured in a new continuous way. The second limitation is generalizability. This study was conducted in a relatively short learning session, and the knowledge type was declarative knowledge with only students from a primary school in China. It was unclear whether the effect of pre-learning positive emotion could last for longer instructional videos. In addition, when it comes to learning procedural knowledge, it may produce different learning results. Finally, do emotions change differently in different cultural contexts for people of different age? Additional studies are needed to examine the efficacy of funny videos to regulate students’ pre-learning emotions in a more extended learning session for different knowledge types or in different cultures. The third research limitation is the type of emotion-inducing videos. The video used in this paper was a funny video, which makes students produce positive emotions. However, there are several kinds of videos that can elicit positive emotions such as food videos and singing and dancing videos. Whether these other kinds of videos can produce the same results as this study remains to be further studied.

## 6. Conclusions

This study examined whether watching funny videos before learning can make students’ emotions more positive, thereby improving their motivation, satisfaction, and task performance. The findings established that students who watched the funny video actually had more positive pre-learning emotions than students who did not. However, no significant differences were reported in learning motivation, learning satisfaction, and knowledge retention between the two groups. Furthermore, students in the experimental group performed better on the transfer test than those in the control group.

## Figures and Tables

**Figure 1 ijerph-19-04434-f001:**
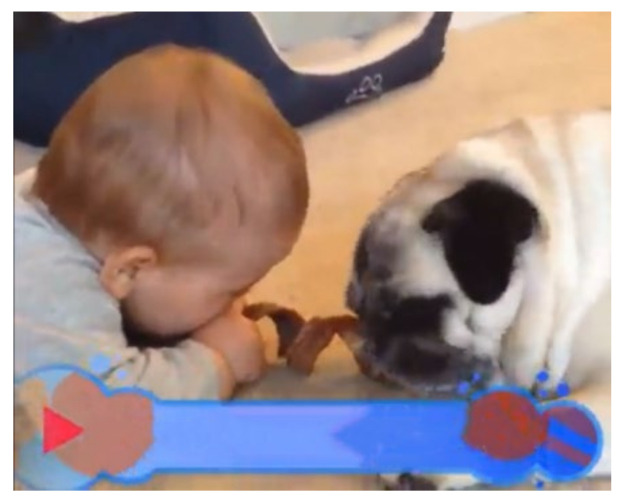
A screenshot of the positive emotion induction video.

**Figure 2 ijerph-19-04434-f002:**
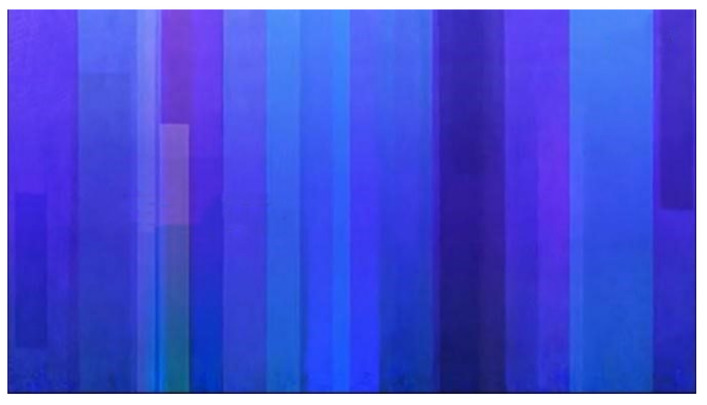
A screenshot of the natural emotion induction video.

**Figure 3 ijerph-19-04434-f003:**
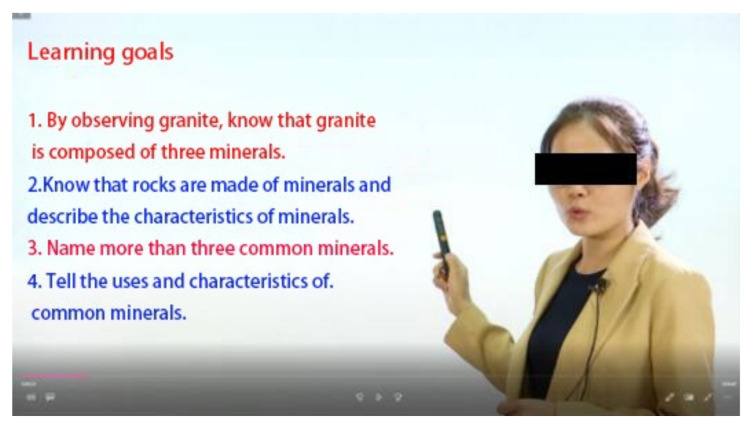
A screenshot of the video lecture.

**Figure 4 ijerph-19-04434-f004:**
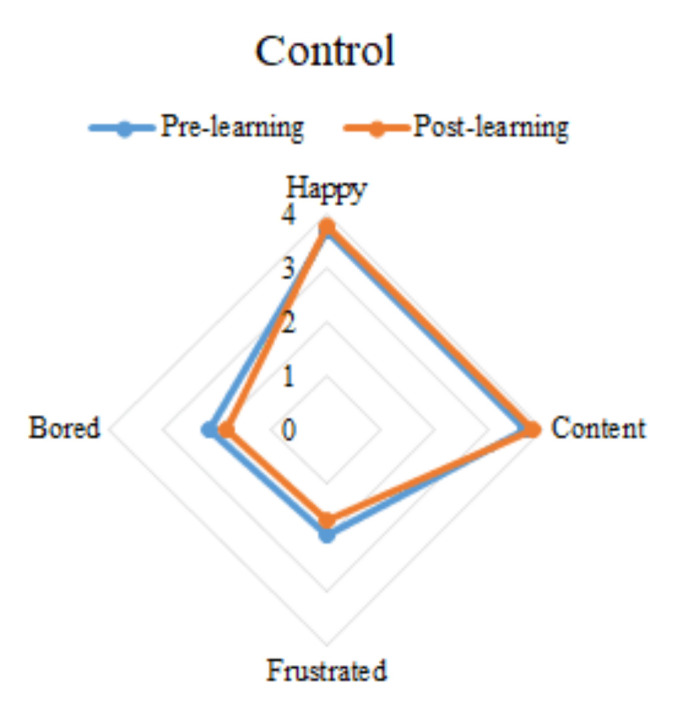
The emotion changes of the control group.

**Figure 5 ijerph-19-04434-f005:**
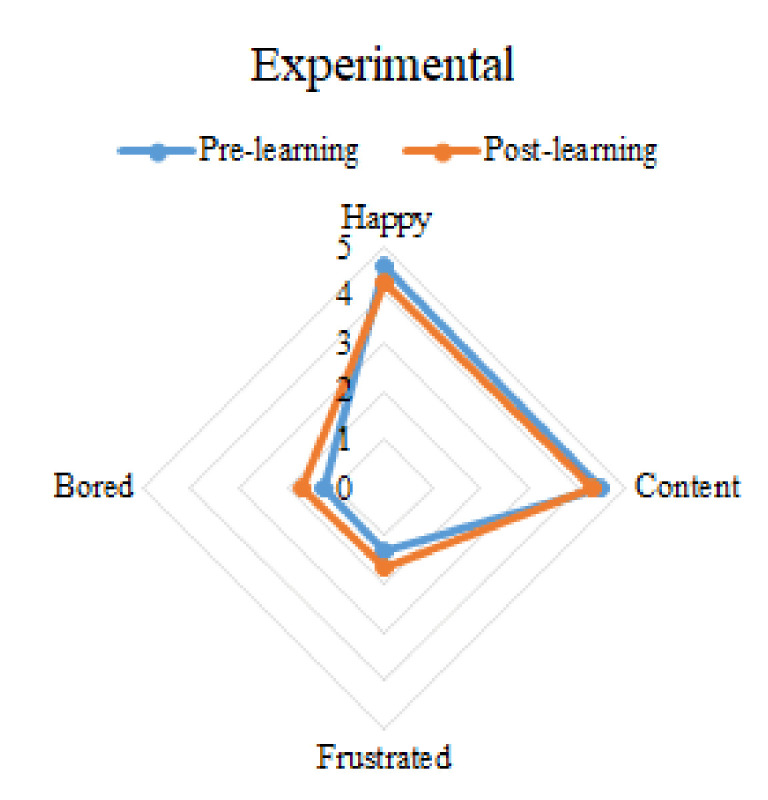
The emotion changes of the experimental group.

**Figure 6 ijerph-19-04434-f006:**
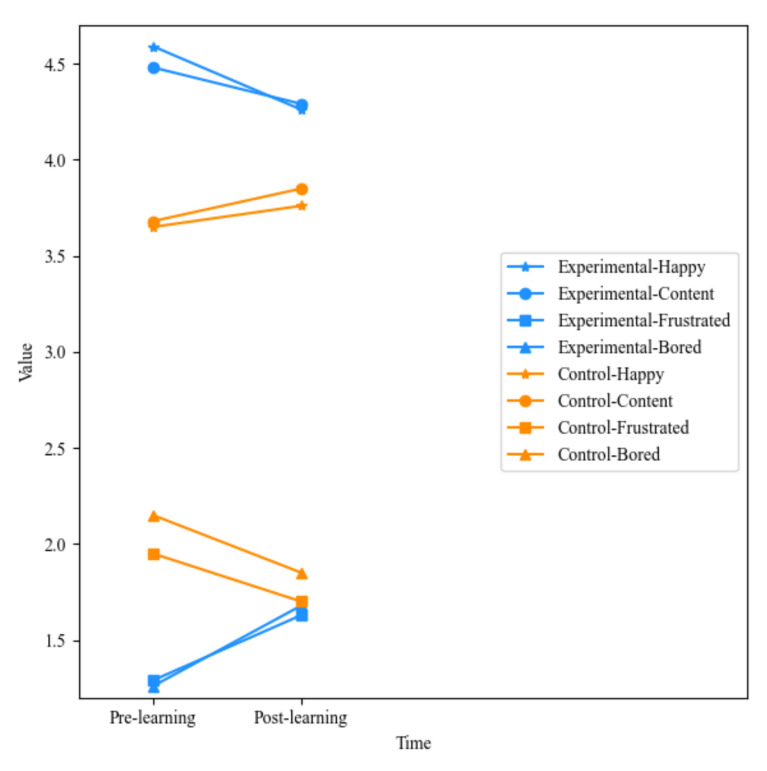
The overall emotion changes of different group at different time points.

**Table 1 ijerph-19-04434-t001:** QIC for different correlation structures.

Correlation Structures	Happy	Content	Frustrated	Bored
Independence	201.892	195.637	167.500	217.864
AR (1)	200.810	195.451	167.422	214.315
Exchangeable	200.810	195.451	167.422	214.315
M-dependent	200.810	195.451	167.422	214.315
Unstructured	200.810	195.451	167.422	214.315

IBM SPSS (version 25, IBM Corporation, Armonk, NY, USA) was used to process the data.

**Table 2 ijerph-19-04434-t002:** The basic characteristics of the two groups.

Items	Control Group	Experimental Group	T	Sig.
Sample size	40	41	/	/
Sex	22 boys, 18 girls	23 boys, 18 girls	/	/
Age	M = 11.12, SD = 0.331	M = 11.20, SD = 0.401	−0.900	0.371
Prior knowledge	M = 8.63, SD = 2.727	M = 8.88, SD = 3.043	0.382	0.703

**Table 3 ijerph-19-04434-t003:** The emotional state of the two groups before learning.

Items	Control Group		Experimental Group		T	Sig.
	M	SD	M	SD		
Pre-happy	3.65	1.442	4.59	0.549	3.809	*p* < 0.001 **
Pre-content	3.68	1.347	4.49	0.790	3.258	0.002 *
Pre-frustrate	1.95	1.218	1.29	0.565	−3.044	0.003 *
Pre-bored	2.15	1.528	1.26	0.503	−3.405	0.001 *

Note. Pre indicated before learning. * *p* < 0.05, ** *p* < 0.001, two-tailed.

**Table 4 ijerph-19-04434-t004:** The tests of model effect.

Independent Variable	Happy		Content		Frustrated		Bored	
	Wald χ^2^	Sig.	Wald χ^2^	Sig.	Wald χ^2^	Sig.	Wald χ^2^	Sig.
Time	0.758	0.384	0.007 *	0.935	0.114	0.736	0.140	0.708
Group	11.098	0.001 *	9.570	0.002 *	3.955	0.047 *	6.635	0.010 *
Time *group	3.127	0.077	1.857	0.173	5.343	0.021 *	5.677	0.017 *

Note. * *p* < 0.05, two-tailed.

**Table 5 ijerph-19-04434-t005:** Multivariable GEE results.

Independent Variables			Happy		Content		Frustrated		Bored	
			M (SD)	Sig.	M (SD)	Sig.	M (SD)	Sig.	M (SD)	Sig.
Time	Pre-learning	Experimental	4.59 (0.086)	<0.001 **	4.48 (0.124)	0.001 *	1.29 (0.090)	0.002 *	1.26 (0.081)	<0.001 **
		Control	3.65 (0.225)		3.68 (0.210)		1.95 (0.190)		2.15 (0.239)	
	Post-learning	Experimental	4.26 (0.118)	0.049 *	4.29 (0.125)	0.071	1.63 (0.161)	0.752	1.68 (0.152)	0.496
		Control	3.76 (0.226)		3.85 (0.208)		1.70 (0.174)		1.85 (0.208)	
Group	Experimental	Pre-learning	4.59 (0.086)	0.004 *	4.48 (0.124)	0.119	1.29 (0.090)	0.007 *	1.26 (0.081)	0.003 *
		Post-learning	4.26 (0.118)		4.29 (0.125)		1.63 (0.161)		1.68 (0.152)	
	Control	Pre-learning	3.65 (0.225)	0.615	3.68 (0.210)	0.470	1.95 (0.190)	0.256	2.15 (0.239)	0.258
		Post-learning	3.76 (0.226)		3.85 (0.208)		1.70 (0.174)		1.85 (0.208)	

Note. * *p* < 0.05, two-tailed. ** *p* < 0.001, two-tailed.

**Table 6 ijerph-19-04434-t006:** Dependent variables between the two groups.

Items	Control Group		Experimental Group		T	Sig.
	M	SD	M	SD		
Motivation	28.78	10.511	31.10	8.958	1.074	0.286
Satisfaction	10.73	6.391	11.44	2.846	0.647	0.520

**Table 7 ijerph-19-04434-t007:** Dependent variables between the two groups.

Items	Control Group		Experimental Group		T	Sig.
	M	SD	M	SD		
Retention	10.61	6.877	12.49	4.325	1.480	0.143
Transfer	5.63	3.794	8.17	2.783	3.437	*p* < 0.001 **

Note: ** *p* < 0.001, two-tailed.

## Data Availability

The data are not publicly available due to privacy restrictions.

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
