# Peer review of "Laugh before You Study: Does Watching Funny Videos before Study Facilitate Learning?"

_ijerph, 2022, doi:10.3390/ijerph19084434_

Round 1
Reviewer 1 Report
This article is very interesting and I read it with joy as this study's primary focus was whether playing funny videos before class can promote students' positive emotions, and then promote students' motivation, satisfaction, and learning outcomes? I have a few suggestions.
- Why the topic is important must be addressed at the start of the introduction.
- The degrees of emotions may vary in different cultures. Where this study was conducted? Does it fit all the nature? Does the result get variation if conducted somewhere else?
- Line 73, explain each ref, and don't just cite it.
- Discuss the future studies in the conclusion.
- Does this study have any limitations? Please discuss.
- What if the nature of videos is increased to more categories? Will that make sense?
- Try to reduce similarity and make it less than 19%.
Author Response
Dear reviewer:
Thanks for your valuable comments and quick responses concerning our manuscript entitled " Laugh Before You Study: Does Watching Funny Video Before Study Facilitate Learning?" (ID: ijerph--1636908). They are very helpful for the improvement of paper quality. We have studied the comments carefully and made corrections seriously according to your suggestions.
Please see the attachment.
Best regards,
The Authors.

Reviewer 2 Report
Dear Authors; This is an interesting study investigating whether playing funny videos before class can promote students' positive emotions, motivation, satisfaction and learning outcomes. However, I find its quality 3 degrees under journal standards. It needs "some serious" work. Regards.
P.S.
[1] Writing
1-1 References: Please "clean up" them according MDPI format. For example, the article years must be bold. There are alot of irregularities.
1-2 Abbreviations: Please add a list of abbreviations right before reference section used in the paper for the readers easy referral. Example: Abbreviations ND: Neural Design; ....
1-3 Discussion & Conclusion Sections: The current status is substandard. Please rewrite these as follows.
5. Discussion.
5.1. This Work
5.1.1. RQ1 title
5.1.2. RQ2 title
5.1.3. RQ3.title
5.1.4. RQ4. title
5.2. Theoretical and Practical Contribution
5.3. Limitations and Future Research
6. Conclusion
Only have a paragraph of 5-6 lines for the conclusion.
[2] Statistical
2-1 Table.1. Basic characteristics of the Participants Some information on the participants age, gender and size of ctl and exp groups are missing. Add them to Table 1. in line 304.
2-2 Randomization Process: In line 279 there are missing information on the randomization process. How it was done ? Add them.
2-3 Missing Statistical Analysis information: Add a new section "3.5. Statistical Analysis" with information of the math formulas for your null hypothesis and alternative hypothesis.
2-4 Missing Statistical Package: Which Statistical Package did you use to analysis your data ? add its name in 3.5. Statistical Analysis and add its relation citation in the reference section.
2-5 Statistical Limitation: The authors applied repeated measure ANOVA in this study. However, this method has some issues and in recent literature "Ordinal Pattern Analysis" has replaced it. The authors are recommended to add this limitation to the study limitations.
Citation: Grice JW, Craig DPA, Abramson CI. A Simple and Transparent Alternative to Repeated Measures ANOVA. SAGE Open. July 2015. doi:10.1177/2158244015604192 https://journals.sagepub.com/doi/pdf/10.1177/2158244015604192
Author Response
Dear editor and reviewers:
Thanks for your valuable comments and quick responses concerning our manuscript entitled " Laugh Before You Study: Does Watching Funny Video Before Study Facilitate Learning?" (ID: ijerph--1636908). They are very helpful for the improvement of paper quality. We have studied the comments carefully and made corrections seriously according to your suggestions.
Please see the attachment.
Best Regards,
The Authors
March 27, 2022

Reviewer 3 Report
Good day and congratulations on the completion of your research project. Your manuscript describes a potentially impactful description of the effect of watching positive videos prior to learning, including the effects on knowledge retention. I found that your study design matches well with the stated aims of your study, and that your methods were described in sufficient detail to allow for independent reproduction.
Lines 242–251: Please define "knowledge retention" and "knowledge transfer", and indicate specifically how the questions used for these test "knowledge retention" and "knowledge transfer", because this is unclear. Please consider providing the questionnaires as supplementary material with your manuscript.
Lines 245–246: The text describes "multiple-choice questions" twice. Please revise.
Line 276: Do you intend to mean "learning satisfaction" here instead of "learning motivation" when describing Cronbach's alpha value?
Table 2 and line 310: Please provide a precise p-value if possible, because p=.000 (or absolute zero) is not a valid probability. Consider using scientific notation (such as p=1 x 10-4) or ranges (such as p<.001) if necessary.
Figure 4–7: If representing results from the Likert-scale happiness questionnaire as parametric data, please include standard deviation along with the mean in Figure 4–7. Also, carefully consider whether your Likert-scale results are parametric in nature, ideally by describing conformance to a normal distribution curve using a statistical test such as the Shapiro–Wilk test. Also, consider combining Figures 4–7 into a single Figure by using a "Radar chart", sometimes called a "spider chart":
https://en.wikipedia.org/wiki/Radar_chart
Line 382: Please define "pre-lass". Do you intend to mean "pre-lesson"?
Please note, your manuscript would benefit from English-language editing for improved clarity, because issues with the language are negatively impacting reader comprehension.
The primary issue is a lack of a definition of "knowledge retention" and "knowledge transfer", including a description of how your questionnaire assesses these attributes. Your methods need to clearly describe and justify your approach to the assessment of these attributes.
This is the end of my recommendations.
Author Response
Dear reviewer:
Thanks for your valuable comments and quick responses concerning our manuscript entitled " Laugh Before You Study: Does Watching Funny Video Before Study Facilitate Learning?" (ID: ijerph--1636908). They are very helpful for the improvement of paper quality. We have studied the comments carefully and made corrections seriously according to your suggestions.
Please see the attachment.
Best Regards,
The Authors
March 27, 2022

Round 2
Reviewer 1 Report
The paper can be accepted now. it is well revised.
Reviewer 2 Report
Dear Authors; most of my concerns were addressed satisfactorily. Regards.